# An analysis of Chinese nursing electronic medical records to predict violence in psychiatric inpatients using text mining and machine learning techniques

Ya-Han Hu [1,2], Jeng-Hsiu Hung [3,4], Li-Yu Hu [5,6], Sheng-Yun Huang [7], Cheng-Che Shen [5,7,8,9] *

1 Department of Information Management, National Central University, Taoyuan City, Taiwan, 2 Asian Institute for Impact Measurement and Management, National Central University, Taoyuan City, Taiwan, 3 Department of Obstetrics and Gynecology, Taipei Tzu Chi Hospital, Buddhist Tzu Chi Medical Foundation, Taipei, Taiwan, 4 School of Medicine, Tzu Chi University, Hualien, Taiwan, 5 School of Medicine, National Yang Ming Chiao Tung University, Taipei, Taiwan, 6 Department of Psychiatry, Taipei Veterans General Hospital, Taipei, Taiwan, 7 Department of Psychiatry, Chiayi Branch, Taichung Veterans General Hospital, Chiayi, Taiwan, 8 Center for Innovative Research on Aging Society (CIRAS), National Chung Cheng University, Minxiong, Taiwan, 9 Department of Post-Baccalaureate Medicine, College of Medicine, National Chung Hsing University, Taichung, Taiwan

* pures1000@yahoo.com.tw

**Data Availability Statement:** All relevant data are within the paper and its Supporting information files.

## Abstract

### Background

The prevalence of violence in acute psychiatric wards is a critical concern. According to a meta-analysis investigating violence in psychiatric inpatient units, researchers estimated that approximately 17% of inpatients commit one or more acts of violence during their stay. Inpatient violence negatively affects health-care providers and patients and may contribute to high staff turnover. Therefore, predicting which psychiatric inpatients will commit violence is of considerable clinical significance.

### Objective

The present study aimed to estimate the violence rate for psychiatric inpatients and establish a predictive model for violence in psychiatric inpatients.

### Methods

We collected the structured and unstructured data from Chinese nursing electronic medical records (EMRs) for the violence prediction. The data was obtained from the psychiatry department of a regional hospital in southern Taiwan, covering the period between January 2008 and December 2018. Several text mining and machine learning techniques were employed to analyze the data.

### Results

The results demonstrated that the rate of violence in psychiatric inpatients is 19.7%. The patients with violence in psychiatric wards were generally younger, had a more violent

**Funding:** This study was partially supported by the Ministry of Science and Technology (grant numbers MOST 108-2314-B-367-001 and 111-2314-B-367-001-MY3). The funders had no role in study design, data collection and analysis, decision to publish, or preparation of the manuscript.

**Competing interests:** The authors have declared that no competing interests exist.

history, and were more likely to be unmarried. Furthermore, our study supported the feasibility of predicting aggressive incidents in psychiatric wards by using nursing EMRs and the proposed method can be incorporated into routine clinical practice to enable early prediction of inpatient violence.

## Conclusions

Our findings may provide clinicians with a new basis for judgment of the risk of violence in psychiatric wards.

---

## Background

People with mental illness are at greater risk of violence, although most do not act violently [1–3]. According to the National Institute of Mental Health's Epidemiologic Catchment Area survey, patients with serious mental illness, including major depressive disorder, schizophrenia, or bipolar disorder, are 2 to 3 times more likely than people without such illnesses to be assaultive [3]. In addition, violence is a common problem that is a frequent cause of injuries to clinicians in psychiatric inpatient units [4, 5]. In a meta-analysis investigating violence in psychiatric inpatient units, the researchers determined that 17% of inpatients committed one or more act of violence during their stay [5]. Inpatient aggression can negatively affect healthcare providers, patients, and the therapeutic environment due to the influence of aggression and the measures implemented to prevent aggression generally being counter-therapeutic [5–7].

A multitude of studies have been conducted on aggression/violence in psychiatric inpatient units [4, 5, 8–30]. Most have focused on individual patient risk factors for aggression/violence. The factors considered to be most associated with patient aggression/violence are the existence of previous episodes, the victim and aggressor being of the same sex, being hostile and impulsive, having experienced involuntary admission, and having longer hospitalization [9, 21, 22]. Furthermore, some studies have revealed situational, relational, and environmental factors to also be related to aggression/violence in inpatient settings [10–12, 22]. A summary of recent studies on the prevalence and risk factors of violence among psychiatric inpatients is presented in Table 1. When the results of studies that recruited different populations were compared in a meta-analysis, however, only a few of the aforementioned factors were determined to be effective predictors of aggression [13]. Predicting violent incidents can be highly challenging [31]. For example, a study demonstrated that inaccuracy in violence risk assessment is often experience-related [32], and Eaton et al indicated that violence may be impossible to predict in patients with mental disorders [33].

Studies have developed a variety of risk assessment measures to improve violence risk assessment [34–43]. However, their performances in different locations vary considerably [44]. In addition, the process of applying some risk assessment measures is cumbersome and time-consuming, and frequently assessing the risk of violence is thus impractical in most real-world clinical settings [45, 46]. Because of these challenges, developing a means through which violent incidents can be predicted by analyzing already-registered clinical text would be a valuable contribution to the field of personalized medicine and would yield time savings.

Most medical institutions use electronic medical record (EMR) systems, and many studies have retrieved unstructured text data from EMRs to investigate various research topics [20, 47–52]. According to the results of our literature review, only two studies used EMRs to predict the risk of aggression in psychiatric inpatients [20, 26], and no studies have used Chinese

**Table 1. Comparative analysis of prevalence and risk factors of violence in psychiatric inpatients: A review of recent studies (2019–2022).**

| Author | Study design | Sample (N) | Definition of aggression/violence | Prevalence | Risk factors |
|---|---|---|---|---|---|
| Brown et al, 2019 (UK) [25] | Retrospective | 394 | With intention to attempt, threaten or inflict harm on another human | 42.2% | A history of head injury |
| Menger et al, 2019 (Netherlands) [26] | Retrospective | Site 1: 3189 Site 2: 3253 | All threatening and violent behavior of a verbal or physical nature directed at another person but excluded self-harm and inappropriate behavior, such as substance use, sexual intimidation, or vandalism | Site 1: 9.1% Site 2: 7.7% | - |
| Girardi et al, 2019 (UK) [27] | Retrospective | 28 | Physical aggression toward others or verbal aggression | 57.2% | Higher scores of Historical Clinical and Risk Management scale |
| Huitema et al, 2021 (Netherlands) [28] | Retrospective | 542 | Verbal aggression, aggression toward objects, self-harm, physical aggression, and sexual aggression | 63.5% | Civil psychiatric patients caused more aggressive incidents than forensic patients and female patients caused more inpatient aggression compared with male patients. |
| Camus et al, 2021 (Switzerland) [21] | Retrospective | 4518 | Violent physical contact directed against another person | 4.4% | Living in sheltered housing before hospitalization; Suffering from schizophrenia with substance abuse comorbidity; Cumulating hospitalization days |
| Fazel et al, 2021 (UK) [29] | Prospective | 89 | An outcome was defined as a violent incident categorized on the Datix system as 'violence' or 'aggression'. | 33% | Total dynamic score of ⏿1; Younger age; Female sex |
| Lockertsen et al, 2021 (Norway) [30] | Retrospective | 528 | Physical violence: a physical act against another person involving the use of body parts or objects, with a clear intention to cause physical injury to that person; Threats as verbal and non-verbal communications conveying a clear intention to inflict physical injury upon another person. | 14.6% | Higher scores of Brøset Violence Checklist |
| McIvor et al, 2022 (UK) [23] | Retrospective | 8923 | - | - | Increased number of violent incidents in the year before admission; Being admitted involuntarily; Being admitted to psychiatric intensive care unit; Instances of self-harm; Being the target of violence; Referral to a Psychiatric Liaison Team |

nursing EMRs to analyze related issues. Therefore, this study established a predictive model for violence in psychiatric inpatients by using structured and unstructured data obtained from Chinese nursing EMRs and several machine learning techniques.

## Methods

### Data sources and study sample

We obtained the data set for the violence prediction task from the psychiatry department of a regional hospital in southern Taiwan. This psychiatry department has 2 25-bed acute psychiatric wards. Admissions from the 2 wards from between January 2008, and December 2018, were included in the data set. The data used in this study were personal data, diagnosis data, and nursing records that were included in EMRs.

### Ethics statement

The current study received approval from the Institutional Review Board (IRB) of Taichung Veterans General Hospital (IRB number: SE19143B). The data set comprised deidentified secondary data. Therefore, the IRB of Taichung Veterans General Hospital formally waived the requirement for participant consent.

### Research variables

The dependent variable was the presence or absence of violence during hospitalization. In this study, we defined violence as any behavior that involves an overt attempt or an actual act of aggression that causes physical harm to another person. This includes behaviors such as threatening to hit or physically attacking someone, damaging property, or throwing objects at people. An example of presence violence in one of nursing EMRs is given below.

> "*After toileting, attempted to climb onto another patient's bed. Despite being reminded of their correct bed location, the patient persisted and even attempted to kick the other patient lying on the bed. Verbal intervention was used to try to stop the patient, but the patient then physically attacked the security staff.*"

Two experienced psychiatrists independently reviewed the nursing EMRs of each inpatient to determine whether violence occurred during the patient's hospitalization. In the case of a discrepancy, a third experienced psychiatrist reviewed the EMRs.

We obtained each patient's personal basic information, admission diagnosis, and admission nursing assessment from the EMRs created by nurses on the first day of hospitalization. After the data were preprocessed, the structured features were combined with the text features extracted by text preprocessing techniques to obtain a complete set of independent variables.

### Text preprocessing of nursing records

Two text preprocessing methods were used to extract text features: bag-of-words (BOW) and sentence embedding.

The BOW method is a commonly used technique in natural language processing (NLP) for text classification and analysis. It involves breaking down a piece of text into individual words, discarding grammar and word order, and representing the text as a numerical vector. Our bag-of-words text preprocessing steps are illustrated in Fig 1. These steps include word transformation, part-of-speech (POS) tagging, and text vectorization.

Initially, we converted full-width characters into half-width characters and removed non-American Standard Code for Information Interchange characters in nursing records.

Next, we used the CkipTagger and NLTK toolkits for text segmentation and POS tagging. Because nursing notes in Taiwan are written bilingually (i.e., in Chinese and English), the text features of each language were separately determined by using different natural language processing (NLP) tools. CkipTagger is an open-source Chinese NLP tool that was released by the Chinese Thesaurus Group (CKIP Lab) of Academia Sinica in 2019. Its main functions include

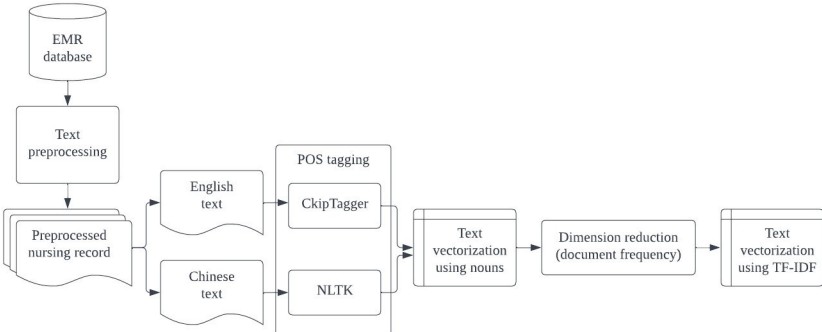

**Fig 1. The text preprocessing steps.**

named entity recognition, word segmentation, and POS tagging. NLTK is one of the most widely used open-source NLP tools for English document preprocessing. Many common text preprocessing tasks can be performed using NLTK, such as tokenization, stemming, POS tagging, and named entity recognition. We first performed word segmentation and POS tagging by using CkipTagger to retrieve a set of Chinese index terms. In this step, we only retained nouns as feature words. NLTK was used for word segments that were marked as foreign words by CkipTagger to identify English index terms.

After the text preprocessing steps were completed, we obtained a large set of feature words from the nursing notes; we employed dimension reduction to filter out meaningless words. The filter was set to a frequency of 0.1 to 0.9 occurrences per document. The term frequency-inverse document frequency (TF-IDF) technique of the Scikit-Learn suite was used for document vectorization.

Sentence embedding is an emerging NLP technique that involves representing a sentence or document as a dense vector of numerical values. The goal of sentence embedding is to capture the meaning and context of a sentence in a low-dimensional space, so that the resulting vector can be used as document features. This study used Sentence-BERT (SBERT) [53], a modification of the Bidirectional Encoder Representations from Transformers (BERT) model, to generate text embedding features. The key innovation of SBERT is the addition of a siamese network architecture, where two copies of the BERT model share the same weights and are trained to encode two different sentences. During training, the model is presented with pairs of sentences and learns to distinguish between sentences that are semantically similar and those that are dissimilar. This enables SBERT to generate high-quality sentence embeddings that capture the semantic meaning of a sentence.

The steps to generate document embeddings using SBERT are as follows: first, split the document into sentences; second, generate sentence embeddings; third, aggregate the sentence embeddings into a single vector by taking the mean average of the sentence embeddings; and finally, normalize the resulting document embedding.

## Descriptive statistics

We performed independent $t$ and chi-squared tests to identify differences between the violent and nonviolent groups.

## Prediction model assessment

We employed 5 well-known supervised learning techniques, namely decision tree (DT, J48 module in WEKA) [54], random forest (RF; RandomForest module in WEKA) [55], support vector machine (SVM, SMO module in WEKA) [56], artificial neural network (ANN, MultilayerPerceptron module in WEKA), K-Nearest Neighbors (KNN, IBk module in WEKA) [57], and boosted random forest (AdaBoost+RF, AdaBoostM1 with RandomForest modules in WEKA) [57], to assess the prediction model performance.

Feature selection is a well-established technique in supervised learning. Feature selection can be employed to improve training efficiency and develop compact models with a high prediction performance. We used the CfsSubsetEval module with the BestFirst search method in Weka, that is, a correlation-based feature selection method, to identify correlations between independent and dependent variables. A feature subset containing features that were highly correlated with the dependent variable but not correlated with each other was obtained.

To improve the prediction performance of the investigated algorithms, the parameter settings can play a crucial role. Therefore, we utilized the CVParameterSelection metalearner module in Weka to optimize these settings. The module allowed us to define multiple

combinations of parameters and automatically execute the base classifier with each combination. It then determined the optimal parameter settings based on the best prediction results obtained through cross-validation.

To mitigate overfitting while building the model, this study employed 10-fold cross-validation process. Firstly, the dataset was randomly and independently divided into 10 subsets. During each iteration, one of these subsets was used as a test dataset while the remaining nine subsets were used as training datasets. This allowed for a more robust evaluation of the model's performance. Moreover, to prevent class imbalance problems, we employed an undersampling method during the cross-validation process. The SpreadSubsample module in Weka 3.8.3 was used to adjust the instance distribution of 2 classes in our training set. The confusion matrix was used to evaluate the prediction performance of each model (Fig 2). True positive (TP) represents the number of inpatients who are correctly predicted to be violent; True negative (TN) represents the number of inpatients who are correctly predicted to be non-violent; False positive (FP) represents the number of inpatients who are incorrectly predicted to be violent; False negative (FN) represents the number of inpatients who are incorrectly predicted to be non-violent.

Using the information summarized in the confusion matrix, 5 classification performance metrics, including accuracy, precision, recall/sensitivity, f1-measure, and specificity, can be obtained through the following equations [58, 59]:

$$Accuracy = \frac{TP + TN}{TP + TN + FP + FN} \tag{1}$$

$$Precision = \frac{TP}{TP + FP} \tag{2}$$

$$Recall = Sensitivity = \frac{TP}{TP + FN} \tag{3}$$

$$F1 - measure = 2 \cdot \frac{precision \cdot recall}{precision + recall} \tag{4}$$

$$Specificity = \frac{TN}{FP + TN} \tag{5}$$

In addition, we also utilized the area under the curve (AUC) to assess the quality of the models. The AUC value can range from 0 to 1, with a higher score indicating a more favorable performance of the classifier.

| Predicted Classes / Actual Classes | Violence | No violence |
|---|---|---|
| Violence | TP | FN |
| No violence | FP | TN |

Fig 2. Confusion matrix.

### Analytical tools

For data extraction, computation, linkage, and processing, we used Microsoft SQL Server 2005 (Microsoft, Redmond, WA, USA). We used SAS (Version 9.2; SAS Institute Cary, NC, USA) and SPSS (Version 19.0 for Windows; IBM, New York, NY, USA) for all statistical analyses, and $P < .05$ was considered to indicate significance. All supervised learning techniques used to assess the prediction model were implemented using Weka 3.8.2 open-source machine learning software (www.cs.waikato.ac.nz/ml/weka).

## Results

### Baseline data

We obtained the EMRs for 2357 inpatients; 293 were excluded because they contained too little text. Finally, 2064 admission records were included. The 5 most common admission diagnoses were schizophrenic disorders [750 cases, 36.3%, *International Classification of Disease*, *ninth revision*, *Clinical Modification* (ICD-9-CM): 295)], episodic mood disorders (527 cases, 25.5%, ICD-9-CM: 296), persistent mental disorders resulting from conditions classified elsewhere (125 cases, 6.1%, ICD-9-CM: 294), other nonorganic psychoses (119 cases, 5.8%, ICD-9-CM: 298), and dementia (84 cases, 4.1%, ICD-9-CM: 290). In addition, the records of 406 admitted patients (19.7%) stated that violence occurred during their hospitalization. The clinical and demographic variables of the violent and nonviolent groups are listed in Table 2. The violent

**Table 2. Characteristics of patients in violent and nonviolent groups.**

| | Violent group | Non-violent group | *P* values |
|---|---|---|---|
| | N = 406 (%) | N = 1658 (%) | |
| Sex | | | |
| Male | 234 (57.6) | 875 (52.8) | 0.078 |
| Female | 172 (42.4) | 783 (47.2) | |
| Age (years) [a] | 46 (36–59) | 49.5 (41–60) | 0.005* |
| Education level | | | |
| 0 | 70 (17.2) | 260 (15.7) | 0.685 |
| 1–6 years | 92 (22.7) | 388 (23.4) | |
| 7–12 years | 183 (45.1) | 779 (47.0) | |
| ≧ 13 years | 51 (12.6) | 183 (11.0) | |
| Occupation | 375 (92.4) | 1488 (89.7) | 0.317 |
| Marriage | | | 0.015* |
| Unmarried | 205 (50.5) | 704 (42.5) | |
| Married | 93 (22.9) | 456 (27.5) | |
| Divorced | 75 (18.5) | 310 (18.7) | |
| Widowed | 13 (3.2) | 90 (5.4) | |
| Violent history | 262 (64.5) | 790 (47.6) | < .001* |
| History of substance use | 78 (19.2) | 331 (20.0) | 0.440 |
| Diagnoses, N (%) | | | |
| Schizophrenic disorders | 136 (33.5) | 614 (37.0) | 0.037* |
| Episodic mood disorders | 98 (24.1) | 429 (25.9) | |
| Persistent mental disorders due to conditions classified elsewhere | 29 (7.1) | 96 (5.8) | |
| Other nonorganic psychoses | 28 (6.9) | 91 (5.5) | |
| Dementia | 22 (5.4) | 62 (3.7) | |

[a] Median (interquartile range);

*Statistical significance

group was younger and had a more violent history than the nonviolent group. Furthermore, patients in the violent group were more likely to be unmarried than those in the nonviolent group were.

After data preprocessing, 3 types of variables were obtained, namely structured variables, TF-IDF document vector variables, and SBERT document embedding variables. For the complete data set, 1032 variables (i.e., 403 structured variables, 245 TF-IDF document vector variables, and 384 SBERT document embedding variables) were obtained. To evaluate the impact of text features on the experimental results, two feature sets were utilized. These sets consisted of structured variables combined with TF-IDF variables (a total of 648 variables) and structured variables combined with SBERT variables (a total of 787 variables). The application of the CfsSubsetEval method resulted in the retention of only 26 independent variables in the dataset, comprising of 21 structured variables, 4 TF-IDF document vector variables, and 1 SBERT document embedding variable.

### Prediction model performance

The results on the performance of the prediction models on the training and validation data sets are presented in Table 3 and Fig 3. The performance of the AdaBoost+RF model was consistently better than that of the other techniques. The structured variables combined with TF-IDF variables resulted in an accuracy of 0.617, a precision of 0.619, a recall/sensitivity of 0.608, an f1-measure of 0.614, a specificity of 0.626, and an AUC of 0.634. The ANN model had the highest sensitivity (0.692) and f1-measure and the KNN model had the highest specificity (0.650). The structured variables combined with SBERT variables resulted in an accuracy of 0.555, a precision of 0.557, a recall/sensitivity of 0.544, an f1-measure of 0.550, a specificity

**Table 3. Prediction model performance assessment using 10-fold cross-validation.**

| Feature set | Method | Metrics | | | | | |
|---|---|---|---|---|---|---|---|
| | | ACC | PRE | SEN | F1 | SPE | AUC |
| Structured + TF-IDF (648) | DT | 0.576 | 0.577 | 0.579 | 0.575 | 0.576 | 0.595 |
| | RF | 0.596 | 0.597 | 0.599 | 0.595 | 0.596 | **0.631** |
| | SVM | **0.601** | **0.604** | 0.589 | 0.596 | 0.613 | 0.601 |
| | KNN | 0.563 | 0.576 | 0.475 | 0.521 | **0.650** | 0.586 |
| | ANN | 0.576 | 0.562 | **0.692** | **0.620** | 0.461 | 0.619 |
| | AdaBoost+RF | 0.617 | 0.619 | 0.608 | 0.614 | 0.626 | 0.634 |
| Structured + SBERT (787) | DT | 0.536 | 0.536 | 0.537 | 0.536 | 0.534 | 0.545 |
| | RF | 0.539 | 0.539 | 0.547 | 0.543 | 0.540 | 0.562 |
| | SVM | 0.546 | 0.547 | 0.534 | 0.540 | 0.557 | 0.546 |
| | KNN | 0.539 | 0.554 | 0.406 | 0.469 | 0.672 | 0.547 |
| | ANN | 0.549 | 0.552 | 0.522 | 0.537 | 0.576 | 0.577 |
| | AdaBoost+RF | 0.555 | 0.557 | 0.544 | 0.550 | 0.567 | 0.587 |
| CfsSubsetEval (26) | DT | 0.555 | 0.558 | 0.534 | 0.546 | 0.576 | 0.581 |
| | RF | 0.639 | **0.650** | 0.603 | 0.626 | 0.675 | 0.677 |
| | SVM | 0.615 | 0.631 | 0.552 | 0.589 | **0.677** | 0.615 |
| | KNN | 0.602 | 0.614 | 0.549 | 0.580 | 0.655 | 0.636 |
| | ANN | 0.584 | 0.586 | 0.569 | 0.578 | 0.599 | 0.600 |
| | AdaBoost+RF | **0.639** | 0.648 | **0.611** | **0.629** | 0.667 | **0.684** |

ACC: accuracy; PRE: precision; SEN: sensitivity/recall; F1: f1-measure; SPE: specificity; AUC: area under the ROC curve; DT: decision tree; RF: random forest; SVM: support vector machine; KNN: *k*-nearest neighbors; ANN: artificial neural network; AdaBoost+RF: AdaBoostM1 with random forest.

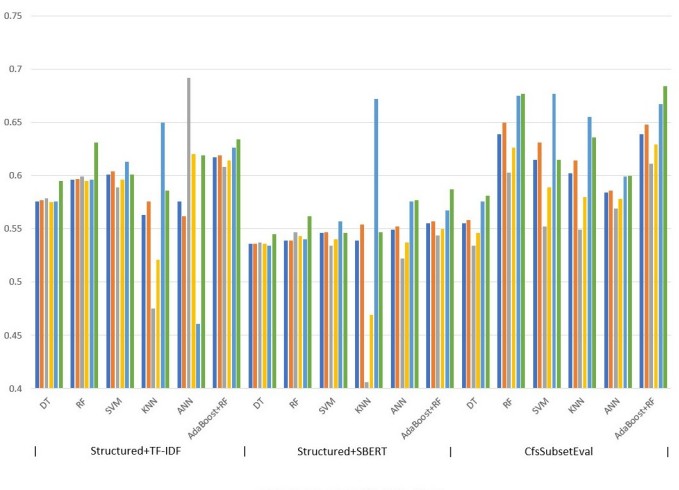

**Fig 3. Prediction model performance assessment using 10-fold cross-validation.**

of 0.567, and an AUC of 0.587. Compared to the use of bag-of-word features, the utilization of sentence embedding variables resulted in poorer performance. In addition, employing feature selection (CfsSubsetEval) resulted in an improved prediction performance for the reduced dataset. When this dataset was fed to AdaBoost+RF, the accuracy improved to 0.639, the precision to 0.648, the f1-measure to 0.629, the specificity to 0.667 and the AUC to 0.684.

## Discussion

Our study is the first to analyze Chinese-language EMRs to predict the risk of violence in psychiatric inpatients by using structured and unstructured data from Chinese EMRs and machine learning techniques. The following are the main findings of our study: (1) the violence rate for psychiatric inpatients was 19.7%; (2) patients in the psychiatric wards with violence were younger, had a more violent history, and were more likely to be unmarried than patients without violence were; (3) violence in psychiatric inpatients could be predicted by using data from Chinese nursing EMRs with an acceptable accuracy (AUC: 0.684).

Patients directing violence at staff or other patients is a common occurrence in most psychiatric treatment facilities. Iozzino et al [5] investigated the prevalence of violent incidents occurring during admission in 35 facilities and determined it to be 2% to 44%, with an average of 17% across the included sites. In our study, the data of 406 patients (19.7%) were determined to indicate that violence occurred during hospitalization. Therefore, the prevalence in the present study was similar with that of Iozzion et al. However, findings regarding the prevalence of violence during admission vary considerably in the literature. This may have occurred for several reasons. First, the definitions of aggression and violence used in studies on psychiatric inpatients can vary considerably, which may contribute to discrepancies in findings regarding the prevalence of violent incidents. For example, in Menger et al, violence was defined as either physical or verbal aggression toward hospital staff or other patients [20]. However, in Lam et al, only physical aggression was considered [16]. In addition, in Schlup et al, psychiatric inpatient violence was categorized and assessed as the following five types: (1) verbal violence; (2) verbal sexual violence; (3) violence against property; (4) physical sexual violence; and (5) physical violence [24]. As Mierlo et al. said, a uniform overall accepted definition of aggressive or violent behavior is lacking, which can result in different operationalizations [60]. It is

important to note that aggression and violence are two distinct concepts. Aggression generally refers to behavior intended to cause harm, whether physical or psychological, while violence specifically involves physical harm, such as hitting, punching, or using a weapon. Violence can be seen as an extreme form of aggression with the primary goal of intentional injury. Therefore, the use of different definitions of violence and aggression in studies can lead to confusion in the interpretation of results. Second, the prevalence of inpatient violence may differ with the type of psychiatric ward (e.g., acute, chronic, forensic, or psychiatric intensive care unit). Third, ethnicity may influence the prevalence of inpatient violence. In their meta-analysis, Dack et al [13] reported no significant ethnicity-related differences between aggressive and nonaggressive patients. However, this result was statistically heterogeneous. Future studies should investigate the relationship between ethnicity and inpatient violence. Our study revealed the rate of violence in Asian acute psychiatric inpatients to be 19.7%.

Consistent with those of other studies [9, 13–19], our findings indicated that a younger age [14, 15, 29], history of violence [16, 17, 23], and unmarried status [18, 19] are related to patients demonstrating violence in psychiatric wards. In a review by Cornaggia et al [9], the researchers concluded that being admitted involuntarily, the existence of previous aggressive episodes, having a longer hospitalization stay, being impulsive, misusing drugs or alcohol, having a younger age, and having a diagnosis of psychosis were associated with inpatient aggression/violence. Furthermore, a review by Dack et al [13] discovered inpatient aggression to be associated with being male, having a younger age, not being married, being admitted involuntarily, having more previous admissions, having a history of exhibiting self-destructive behavior, having a history of substance misuse, and having a history of violence.

Research indicates that predicting violence incidents can be challenging and is often experience-related [31, 32]. Therefore, several risk assessment instruments have been developed [61]. The most commonly used risk assessment instruments are the Violence Risk Appraisal Guide [39], Structured Assessment of Aggression Risk in Youth [41], and Historical Clinical Risk Management-20 [40]. Sing et al [62] determined in a meta-study that the aforementioned instruments have median AUCs of between 0.70 and 0.74. However, the effects identified using these instruments have generally been small. In addition, the studies that have employed these instruments have mostly had heterogeneous patient populations and differing reports regarding the performance of the instruments. These factors prevent the instruments' predictive abilities from being generalized to other facilities [13, 44]. In addition, the application of some risk assessment instruments is time-consuming, which has rendered their frequent use in most real-world clinical settings impractical [45, 46]. Because of the aforementioned challenges, using clinical text that is already registered in EMRs to predict violent incidents may be a practical method for violence risk assessment. In our study, violence in psychiatric inpatients could be predicted using data in Chinese nursing EMRs with an AUC of 0.684.

The results of the 10-fold cross-validation revealed AdaBoost+RF to have the highest average AUC. RF has been reported to have a more favorable performance than many other conventional supervised learning techniques in numerous studies [63–66]. RF offers the advantages of not assuming a linear relationship in the model; employing ensemble learning, in which a strong learner is formed by combining weak learner groups; and iteratively sampling data and completing embedded feature selection to create several decision trees. On the other hand, boosting (i.e., AdaBoostM1 in WEKA) is a technique used to improve the performance of weak learners by weighting misclassified observations and re-training the model to focus more on these observations in the subsequent iterations. This allows the model to gradually improve its performance by focusing on the data points that are more difficult to classify.

Additionally, boosting can also help RF to capture more diverse and informative features by allowing each tree to focus on different subsets of the data. By combining the benefits of both techniques, AdaBoost+RF can improve prediction performance and is less prone to overfitting.

Through a literature review, we discovered that only 2 studies, that of Menger et al, has applied machine learning techniques to EMR data to predict the risk of aggression in psychiatric inpatients [20]. Menger et al achieved the highest accuracy when they combined Document Embeddings with a Recurrent Neural Network. The AUC of our study (AUC: 0.684) is lower than that obtained in Menger et al (AUC: 0.764–0.797). The differences in the study designs may be responsible for the discrepancy. Menger et al investigated incidents in which patients demonstrated either physical or verbal aggression toward staff or other patients [20]. However, only physical aggression was included in our study. Furthermore, Menger et al included the EMRs of doctors and nurses that were created on the first day of admission, whereas our study included only the EMRs of nurses created on the first day of admission. Including less data in analyses can result in less accurate predictions. In addition, Chinese is a complex language with multiple meanings for the same word, which can make it difficult for text mining algorithms to accurately identify the intended meaning of the text [67–69]. Medical terminology in Chinese can be ambiguous, with different terms used to describe similar symptoms or conditions. This ambiguity can cause confusion for text mining algorithms, leading to inaccurate or incomplete results. Furthermore, Chinese EMRs may not be standardized, meaning there is a lack of consistency in how patient information is recorded. This can make it difficult for text mining algorithms to accurately detect violence from the EMR data.

## Limitations

In this study, we used text mining of Chinese EMRs to predict violence in psychiatric inpatients. While our results provide some insights into the potential of this approach, there are several limitations to our study that should be discussed. First, the use of text mining to extract data from EMRs may not capture all relevant information on risk factors for violence in psychiatric inpatients. For example, data on whether admission was voluntary, the number of previous admissions, quality of life, family support, intelligence level, and history of sexual abuse, which are known to be associated with the risk of violence in psychiatric wards [4, 5, 8–19], were not included in our data set. As a result, the accuracy of our predictions may be limited by the absence of these important factors. Second, the accuracy of our predictions may also be limited by the quality and completeness of the EMRs used in our study. It is possible that errors or inconsistencies in documentation could have affected the reliability of our data and our ability to accurately identify risk factors for violence. Third, the use of Chinese EMRs may introduce cultural and linguistic biases that could impact the validity of our predictions. It is important to acknowledge that the cultural and linguistic factors that may influence violence in psychiatric inpatients are complex and may not be fully captured by our text mining approach. Fourth, the EMRs did not clearly specify the methods through which diagnoses were made. Therefore, we could not evaluate the diagnostic accuracy of the psychiatric disorders reported in the EMRs included in our study. Fifth, our study did not include verbal aggression. Therefore, the prevalence of verbal aggression in psychiatric wards and the accuracy of verbal aggression prediction in psychiatric wards using Chinese EMRs warrant further study. Finally, the timeframe of our study may have impacted the accuracy of our predictions, as violence in psychiatric inpatients could be influenced by factors that change over time, such as changes in medication or therapy.

## Conclusion and future directions

Violence is a key concern in acute psychiatric wards because it can lead to patient or staff injury and because it is counter-therapeutic. Studies have reported that 75% to 100% of nursing staff who work in acute psychiatric units have experienced patient assault [70, 71]. Aggression toward staff was indicated to contribute to high staff turnover [72]. Given the importance of this problem, predicting which psychiatric inpatients will commit violence is crucial. Therefore, we established a predictive model for violence in psychiatric inpatients by using structured and unstructured data obtained from Chinese EMRs and several machine learning techniques with an acceptable accuracy. The results supported the feasibility of predicting violent incidents in psychiatric wards by using EMR data collected at the time of admission and indicated that such a method might be incorporated into routine clinical practice to enable early prediction of inpatient violence. Our findings may provide clinicians with a new basis for judging violence risk in psychiatric wards and may enable first-line caregivers to implement appropriate treatment and preventive measures for hospitalized patients at high risk of violence, ultimately improving patient outcomes and staff safety.

Future research directions in this field could include incorporating additional variables, such as admission type, previous admissions, intelligence level, and history of sexual abuse, to improve the accuracy of predictive models for violence. Structured interviews could be used to determine psychiatric diagnoses and investigate the association between psychiatric disorders and the risk of violence in inpatients. Future studies could also explore the prevalence of verbal aggression in psychiatric wards and the accuracy of predicting verbal aggression using EMRs. Furthermore, validation of our model on other populations to determine its generalizability and applicability to different contexts is needed. The effectiveness of different machine learning techniques and prediction models could also be compared to identify the most accurate and efficient method for predicting violence in psychiatric inpatients. Moreover, targeted interventions could be developed and implemented to reduce the risk of violence in psychiatric inpatients identified as high-risk by the model. Finally, long-term outcomes of violence in psychiatric inpatients, such as patient outcomes and staff safety, should be examined to determine the impact of early prediction and intervention on patient care and outcomes.

## Supporting information

**S1 Data. CfsSubsetEval (26)-balanced-final.**
(CSV)

**S2 Data. Structured with SBERT (787)-balanced-final.**
(CSV)

**S3 Data. Structured with TF-IDF (648)-final.**
(CSV)

## Author Contributions

**Conceptualization:** Li-Yu Hu.

**Data curation:** Cheng-Che Shen.

**Formal analysis:** Ya-Han Hu.

**Funding acquisition:** Cheng-Che Shen.

**Investigation:** Li-Yu Hu, Cheng-Che Shen.

**Methodology:** Ya-Han Hu.

**Project administration:** Cheng-Che Shen.

**Supervision:** Li-Yu Hu.

**Validation:** Li-Yu Hu.

**Writing – original draft:** Cheng-Che Shen.

**Writing – review & editing:** Jeng-Hsiu Hung, Sheng-Yun Huang.

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
