## [Decision Letter · Decision Letter 0]

13 Feb 2023

PONE-D-23-00580Predicting Aggression in Psychiatric Inpatients Using Text Mining of Chinese Electronic Nursing RecordsPLOS ONE

Dear Dr. Shen,

Thank you for submitting your manuscript to PLOS ONE. After careful consideration, we feel that it has merit but does not fully meet PLOS ONE’s publication criteria as it currently stands. Therefore, we invite you to submit a revised version of the manuscript that addresses the points raised during the review process.

We look forward to receiving your revised manuscript.

Kind regards,

Qin Xiang Ng, MD, MPH

Academic Editor

PLOS ONE

Journal Requirements:

3. Please amend your manuscript to include your abstract after the title page.

Reviewers' comments:

Reviewer's Responses to Questions

**Comments to the Author**

1. Is the manuscript technically sound, and do the data support the conclusions?

Reviewer #1: Yes

Reviewer #2: Partly

2. Has the statistical analysis been performed appropriately and rigorously? 

Reviewer #1: Yes

Reviewer #2: Yes

3. Have the authors made all data underlying the findings in their manuscript fully available?

Reviewer #1: No

Reviewer #2: No

4. Is the manuscript presented in an intelligible fashion and written in standard English?

Reviewer #1: Yes

Reviewer #2: No

5. Review Comments to the Author

Reviewer #1: Thank you for the work to keep the hospitals safe, my interpretation is that this is a technical paper with a test of concept:

- The authors seemed to have used the terms "aggression" and "violence" interchangeably. Researchers generally define violence as aggression intended to cause extreme physical harm (e.g., injury, death). Thus, all violent acts are aggressive, but not all aggression are violent.

- Wouldn't a past history of aggression be the most significant predictor of future aggression?

- Could the authors provide a text sample assessed by the psychiatrists and the software as to what constituted a positive example of "aggression"

- Also wanted to know the rationale behind not including "verbal aggression" in the definition of aggression.

- Could the authors give explanations as to why the text mining failed at detecting aggression (AUC of approximately only 0.65)?

Reviewer #2: In this work, several text mining and machine learning techniques were used to analyze the tested data. The authors estimate the physical aggression rate for psychiatric inpatients and establish a predictive model for aggression in psychiatric inpatients. The paper's contribution to existing knowledge in this research field is not well justified. The authors mentioned some recent techniques, but the paper needs to address the motivation for developing another method. The paper needs to contribute more, and the following points can improve the manuscript.

1. The title can be improved.

2. A comparative study can be added to the BACKGROUND section in table form to show the recent efforts.

3. Figure 1 should be improved.

4. The novelty of this work is not clear. Clarify this.

5. Performance evaluation metrics are not enough. Add some other metrics and explain them mathematically.

6. The proposed method should be compared with more recent techniques.

7. Tabular data should be presented with the graphs.

8. There should be some discussion on the limitations of the methods presented in a separate section.

9. References should be updated; there are no references in 2022.

10. The manuscript organization should be improved.

11. Improve the English of the work. There are too many problems with paper typesetting.

12. Change the “Conclusion” section title to “conclusion and future directions” and add more discussion and future directions to the research.

13. The paper is unsuitable for acceptance in its current form. The article needs rewriting to address the comments mentioned above.

6. PLOS authors have the option to publish the peer review history of their article (what does this mean?). If published, this will include your full peer review and any attached files.

Reviewer #1: No

Reviewer #2: No

---

## [Author Response · Author response to Decision Letter 0]

1 May 2023

Journal Requirements:

Response: Thank you for your message. We have made every effort to ensure that our manuscript meets PLOS ONE's style requirements, including the file naming conventions. If there are any specific areas that need further attention, please let us know and we will be happy to make the necessary revisions.

Response: Thank you for your message and for providing us with information about PLOS ONE's data policy. We apologize for not specifying where the minimal data set underlying the results described in our manuscript can be found in the initial submission. We have carefully reviewed our data and are pleased to inform you that we will upload our study's minimal underlying data set as Supporting Information files upon re-submitting our revised manuscript. We understand the importance of making the minimal data set fully available, and we will ensure that any potentially identifying patient information is fully anonymized. Thank you again for your feedback and for your assistance in ensuring that our manuscript meets the necessary requirements.

3. Please amend your manuscript to include your abstract after the title page.

Response: We have added our abstract after the title page.

Reviewers' comments:

Reviewer #1:

1. Thank you for the work to keep the hospitals safe, my interpretation is that this is a technical paper with a test of concept:

Response: We are very grateful for the reviewers' suggestions to make our article more suitable for publication, and we have done our best to modify the article according to the reviewers' suggestions.

2. The authors seemed to have used the terms "aggression" and "violence" interchangeably. Researchers generally define violence as aggression intended to cause extreme physical harm (e.g., injury, death). Thus, all violent acts are aggressive, but not all aggression are violent.

Response: Thank you for your insightful comments on our manuscript. We appreciate your attention to detail and have carefully reviewed the use of the terms "aggression" and "violence" in our manuscript. As you have correctly pointed out, aggression and violence are two distinct concepts. Aggression is generally considered a behavior that is intended to cause physical or psychological harm to another person. Violence, on the other hand, refers to any behavior that causes physical harm to another person, such as hitting, punching, or using a weapon. Violence is more likely to refer to an extreme form of aggression that has intentional injury as its primary goal. We agree that it is important to distinguish between violence and aggression, particularly in the context of psychiatric inpatients. We will revise our manuscript to clarify the difference between aggression and violence and ensure that these terms are used appropriately throughout the manuscript.

However, we would like to mention that in previous studies related to aggression and violence in psychiatric inpatients, there has been a lack of consistency in the use of these terms. Some studies have used "aggression" as the main topic, while others have used "violence." Moreover, some studies have used both terms without clear definitions, leading to confusion in the interpretation of the results. In our manuscript, when referring to previous studies, we have used the descriptions consistent with the original research papers. Additionally, in the Discussion section of our paper, we will highlight the potential impact of using either "aggression" or "violence" in research, as it may influence the interpretation of results. We appreciate your feedback and will ensure that our manuscript is clear and consistent in its use of these terms.

3. Wouldn't a past history of aggression be the most significant predictor of future aggression?

Response: Thank you for your feedback and suggestion. Indeed, a past history of violence is an important factor in predicting future violent behavior, and it is one of the factors we considered in our study. Consistent with previous studies, our study showed the violent group had a more violent history than the nonviolent group. As the reviewer said, we believe that while a past history of violence is indeed an important factor in predicting future violent behavior. 

4. Could the authors provide a text sample assessed by the psychiatrists and the software as to what constituted a positive example of "aggression"

Response: Thank you for your inquiry regarding a text sample that demonstrates positive examples of violence. As requested, we would like to provide an example of a nursing record that was assessed by the psychiatrists as a positive example of inpatient violence:

"After toileting, the patient attempted to climb onto another patient's bed. Despite being reminded of their correct bed location, the patient persisted and even attempted to kick the other patient lying on the bed. Verbal intervention was used to try to stop the patient, but the patient then physically attacked the security staff." This incident was considered an example of inpatient violence in our study.”

We appreciate your suggestion and have added this example to our manuscript to provide a clearer illustration of the criteria used to define inpatient violence (page 7, lines 105-109).

5. Also wanted to know the rationale behind not including "verbal aggression" in the definition of aggression.

Response: Thank you for your question regarding the exclusion of verbal aggression in our definition of violence. Our rationale behind this decision is that verbal aggression may not always be recorded in nursing records due to its lower severity compared to physical aggression and destruction. Since our study relied on reviewing the content of nursing records to determine the occurrence of violence, including verbal aggression in our definition may lead to an underestimation of its prevalence and consequently affect the predictive outcomes.

6. Could the authors give explanations as to why the text mining failed at detecting aggression (AUC of approximately only 0.65)?

Response: There are several possible reasons why text mining failed at detecting violence. First, Chinese is a complex language with multiple meanings for the same word, which can make it difficult for text mining algorithms to accurately identify the intended meaning of the text. Second, Medical terminology in Chinese can be ambiguous, with different terms used to describe similar symptoms or conditions. This ambiguity can cause confusion for text mining algorithms, leading to inaccurate or incomplete results. Third, Chinese EMRs may not be standardized, meaning there is a lack of consistency in how patient information is recorded. This can make it difficult for text mining algorithms to accurately detect violence from the EMR data. We have included these explanations in the revised version of our manuscript (page 21, lines 345-353).

Reviewer #2:

1. In this work, several text mining and machine learning techniques were used to analyze the tested data. The authors estimate the physical aggression rate for psychiatric inpatients and establish a predictive model for aggression in psychiatric inpatients. The paper's contribution to existing knowledge in this research field is not well justified. The authors mentioned some recent techniques, but the paper needs to address the motivation for developing another method. The paper needs to contribute more, and the following points can improve the manuscript.

Response: We are very grateful for the reviewers' suggestions to make our article more suitable for publication, and we have done our best to modify the article according to the reviewers' suggestions

2. The title can be improved.

Response: We appreciate the reviewer's constructive suggestion for improving our work. We have revised the title of our manuscript to "An Analysis of Chinese Nursing Electronic Medical Records to Predict Violence in Psychiatric Inpatients using Text Mining and Machine Learning Techniques" We believe that this new title accurately reflects the focus and contributions of our research, and we hope it will better engage potential readers.

3. A comparative study can be added to the BACKGROUND section in table form to show the recent efforts.

Response: Thank you for your valuable feedback. We appreciate your suggestion to include a comparative study in table form in the background section to showcase recent efforts in this area. As per your suggestion, we have added a table (Table 1) comparing the prevalence of violence and associated risk factors across various studies conducted in the last three years. The table will help readers understand the similarities and differences among the studies and highlight the research gaps. We hope this addition will enhance the quality of our manuscript, and we thank you once again for your insightful feedback. The proposed title for the table is: "Comparative Analysis of Prevalence and Risk Factors of Violence in Psychiatric Inpatients: A Review of Recent Studies (2019-2022)."

4. Figure 1 should be improved.

Response: Thank you for reviewing our manuscript. We appreciate your feedback regarding Figure 1. We have made significant improvements to Figure 1. We have modified the figure to make it more visually appealing and easier to understand. The revised figure is presented below.

5. The novelty of this work is not clear. Clarify this.

Response: Thank you for your comments on our paper. We appreciate the opportunity to address your concerns.

Regarding the novelty of our work, we believe that our study makes several important contributions. Firstly, we utilized text mining and machine learning techniques to predict violence in psychiatric inpatients based on nursing electronic medical records. To the best of our knowledge, this is one of the first studies to apply these techniques specifically to Chinese nursing electronic medical records.

Secondly, our study expands the current literature on violence prediction in psychiatric inpatients, as previous studies have mainly focused on clinical variables, such as demographics, diagnoses, and medication use. Our study demonstrates the potential of using nursing records, which contain rich behavioral and contextual information, as a valuable source for predicting aggression.

Finally, our study also contributes to the field of mental health care by providing a potential tool for early detection and prevention of violence in psychiatric inpatients. This can improve patient safety and well-being, as well as reduce the burden on mental health care providers.

We hope that this clarifies the novelty of our work. Please let us know if you have any further questions or concerns.

6. Performance evaluation metrics are not enough. Add some other metrics and explain them mathematically.

Response: Thank you for your valuable feedback. We agree that more performance evaluation metrics can provide a more comprehensive understanding of the model's effectiveness. To further enhance our analysis, we include three more metrics such as precision, recall, and f1-score (page 13, lines 198-200). We have also provided equations of the selected metrics (page 13, lines 201-205). For details, please refer to the fifth paragraph of the Prediction Model Assessment section (pages 11-13).

7. The proposed method should be compared with more recent techniques.

Response: In response to this comment, we have included two more recent text mining and machine learning techniques: sentence-BERT and boosted random forest. Sentence-BERT, or SBERT, is a powerful technique for generating text embeddings, which are numerical representations of the meaning and semantic content of a given piece of text. One of the key strengths of SBERT is its ability to capture the nuances of language and the subtleties of meaning that are often lost in other embedding techniques. This is achieved through the use of siamese neural networks, which are trained to encode the meaning of two similar sentences in such a way that their embeddings are very close together in vector space, while the embeddings of dissimilar sentences are far apart. This allows SBERT to generate highly discriminative embeddings that can be used for a wide range of natural language processing tasks, such as text classification, semantic search, and information retrieval. 

Boosted random forest is a powerful machine learning technique that has several advantages over conventional approaches. One of the key strengths of BRF is its ability to handle complex and high-dimensional data, which are often encountered in real-world applications. Boosted random forest achieves this by combining the strengths of two powerful algorithms: Random Forest (RF) and Boosting. RF is a well-known machine learning technique that is based on decision trees and is effective at handling large datasets with many features. Boosting, on the other hand, is a method of combining weak learners into a strong learner, which can improve the overall performance of the model. By combining these two techniques, boosted random forest is able to generate more accurate and robust predictions than conventional machine learning algorithms.

In the experimental evaluation, we found that the models constructed using TF-IDF text features outperformed SBERT. In addition, boosted random forest exhibited higher predictive performance compared to the conventional classification techniques. For details, please refer to the “Prediction Model Performance” section.

8. Tabular data should be presented with the graphs.

Response: We appreciate your feedback regarding the presentation of our data. We have carefully considered your suggestion and have now included tabular data alongside the graphs presented in the manuscript (Table 3 and Figure 3). 

9. There should be some discussion on the limitations of the methods presented in a separate section.

Response: Thank you for your comments on our paper. We appreciate your suggestion regarding the need to discuss the limitations of our methods in a separate section. We have taken this feedback into consideration and have revised our manuscript accordingly. The revised section is presented below.

“In this study, we used text mining of Chinese EMRs to predict violence in psychiatric inpatients. While our results provide some insights into the potential of this approach, there are several limitations to our study that should be discussed. First, the use of text mining to extract data from EMRs may not capture all relevant information on risk factors for violence in psychiatric inpatients. For example, data on whether admission was voluntary, the number of previous admissions, quality of life, family support, intelligence level, and history of sexual abuse, which are known to be associated with the risk of violence in psychiatric wards [4, 5, 8-20], were not included in our data set. As a result, the accuracy of our predictions may be limited by the absence of these important factors. Second, the accuracy of our predictions may also be limited by the quality and completeness of the EMRs used in our study. It is possible that errors or inconsistencies in documentation could have affected the reliability of our data and our ability to accurately identify risk factors for violence. Third, the use of Chinese EMRs may introduce cultural and linguistic biases that could impact the validity of our predictions. It is important to acknowledge that the cultural and linguistic factors that may influence violence in psychiatric inpatients are complex and may not be fully captured by our text mining approach. Fourth, the EMRs did not clearly specify the methods through which diagnoses were made. Therefore, we could not evaluate the diagnostic accuracy of the psychiatric disorders reported in the EMRs included in our study. Fifth, our study did not include verbal aggression. Therefore, the prevalence of verbal aggression in psychiatric wards and the accuracy of verbal aggression prediction in psychiatric wards using Chinese EMRs warrant further study. Finally, the timeframe of our study may have impacted the accuracy of our predictions, as violence in psychiatric inpatients could be influenced by factors that change over time, such as changes in medication or therapy.”

10. References should be updated; there are no references in 2022.

Response: Thank you for your feedback on our manuscript. As per your comment, we have added multiple recent references in the field, such as "Instruments for Measuring Violence on Acute Inpatient Psychiatric Units: Review and Recommendations," published in Psychiatr Serv in June 2022, and "Inpatient violence in a psychiatric hospital in the middle of the pandemic: clinical and community health aspects," published in AIMS Public Health in February 2022. We have also made sure to review and update all our references to ensure they are up-to-date and relevant to our study. We thank you for bringing this to our attention, and we hope that the updated references enhance the quality and rigor of our manuscript.

11. The manuscript organization should be improved.

Response: Thank you for your review of our manuscript. We appreciate your feedback and suggestions for improving the organization of the paper. We have carefully reviewed the manuscript's structure and made necessary changes to ensure that the flow of ideas is clear and logical. We have also reorganized the content to improve its readability and overall coherence. If you have any further suggestions or comments, please do not hesitate to let us know. We are committed to addressing any remaining issues and improving our manuscript to meet your expectations.

12. Improve the English of the work. There are too many problems with paper typesetting.

Response: Thank you for your review of our manuscript. We appreciate your feedback on improving the English of the work and paper typesetting. We take your comments seriously and have worked diligently to improve the language and overall presentation of our manuscript. In addition, we have also sent our manuscript to a professional English editing service to ensure that the language is of the highest quality. We believe that the modifications made based on your feedback and the additional editing from the English editing service have significantly improved the quality and readability of our manuscript. We thank you for bringing these issues to our attention and for helping us to improve our work.

13. Change the “Conclusion” section title to “Conclusion and future directions” and add more discussion and future directions to the research.

Response: Thank you for your valuable feedback on our manuscript. We have made the requested changes to the "Conclusion" section as follows:

We have changed the title of the "Conclusion" section to "Conclusion and Future Directions" to reflect the inclusion of future research directions in this field. In this revised section, we have expanded our discussion of the results and added more future directions for research. The revised section is presented below.

“Violence is a key concern in acute psychiatric wards because it can lead to patient or staff injury and because it is counter-therapeutic. Studies have reported that 75% to 100% of nursing staff who work in acute psychiatric units have experienced patient assault [72, 73]. Aggression toward staff was indicated to contribute to high staff turnover [74]. Given the importance of this problem, predicting which psychiatric inpatients will commit violence is crucial. Therefore, we established a predictive model for aggression violence in psychiatric inpatients by using structured and unstructured data obtained from Chinese EMRs and several machine learning techniques with an acceptable accuracy. The results supported the feasibility of predicting aggressive violent incidents in psychiatric wards by using EMR data collected at the time of admission and indicated that such a method might be incorporated into routine clinical practice to enable early prediction of inpatient violence. Our findings may provide clinicians with a new basis for judging aggression violence risk in psychiatric wards and may enable first-line caregivers to implement appropriate treatment and preventive measures for hospitalized patients at high risk of violence, ultimately improving patient outcomes and staff safety.

Future research directions in this field could include incorporating additional variables, such as admission type, previous admissions, intelligence level, and history of sexual abuse, to improve the accuracy of predictive models for violence. Structured interviews could be used to determine psychiatric diagnoses and investigate the association between psychiatric disorders and the risk of violence in inpatients. Future studies could also explore the prevalence of verbal aggression in psychiatric wards and the accuracy of predicting verbal aggression using EMRs. Furthermore, validation of our model on other populations to determine its generalizability and applicability to different contexts is needed. The effectiveness of different machine learning techniques and prediction models could also be compared to identify the most accurate and efficient method for predicting violence in psychiatric inpatients. Moreover, targeted interventions could be developed and implemented to reduce the risk of violence in psychiatric inpatients identified as high-risk by the model. Finally, long-term outcomes of violence in psychiatric inpatients, such as patient outcomes and staff safety, should be examined to determine the impact of early prediction and intervention on patient care and outcomes.”

14. The paper is unsuitable for acceptance in its current form. The article needs rewriting to address the comments mentioned above.

Response: Thank you for taking the time to review our manuscript and for providing valuable feedback. We appreciate your efforts in helping us improve the quality of our work. We have carefully considered all of your comments and suggestions and have made significant revisions to the manuscript accordingly. We hope that our modifications have addressed your concerns. We believe that the changes we have made have significantly strengthened the manuscript, and we are confident that it now meets the high standards of the journal. Once again, thank you for your help in making our paper better. We hope that our revised manuscript will be acceptable for publication in your esteemed journal.

---

## [Decision Letter · Decision Letter 1]

15 May 2023

An Analysis of Chinese Nursing Electronic Medical Records to Predict violence in Psychiatric Inpatients using Text Mining and Machine Learning Techniques

PONE-D-23-00580R1

Dear Dr. Shen,

We’re pleased to inform you that your manuscript has been judged scientifically suitable for publication and will be formally accepted for publication once it meets all outstanding technical requirements.

Kind regards,

Qin Xiang Ng, MBBS, MPH

Academic Editor

PLOS ONE

Additional Editor Comments (optional):

Reviewers' comments:

Reviewer's Responses to Questions

**Comments to the Author**

1. If the authors have adequately addressed your comments raised in a previous round of review and you feel that this manuscript is now acceptable for publication, you may indicate that here to bypass the “Comments to the Author” section, enter your conflict of interest statement in the “Confidential to Editor” section, and submit your "Accept" recommendation.

Reviewer #1: All comments have been addressed

Reviewer #2: All comments have been addressed

2. Is the manuscript technically sound, and do the data support the conclusions?

Reviewer #1: Yes

Reviewer #2: Partly

3. Has the statistical analysis been performed appropriately and rigorously? 

Reviewer #1: Yes

Reviewer #2: Yes

4. Have the authors made all data underlying the findings in their manuscript fully available?

Reviewer #1: Yes

Reviewer #2: Yes

5. Is the manuscript presented in an intelligible fashion and written in standard English?

Reviewer #1: Yes

Reviewer #2: Yes

6. Review Comments to the Author

Reviewer #1: (No Response)

Reviewer #2: The authors have addressed most of my concerns. The paper can be accepted. It is recommended to have a list of abbreviations in table form in the introduction section.

7. PLOS authors have the option to publish the peer review history of their article (what does this mean?). If published, this will include your full peer review and any attached files.

Reviewer #1: No

Reviewer #2: No

---

## [Editor Report · Acceptance letter]

30 May 2023

PONE-D-23-00580R1 

An Analysis of Chinese Nursing Electronic Medical Records to Predict violence in Psychiatric Inpatients using Text Mining and Machine Learning Techniques 

Dear Dr. Shen:

I'm pleased to inform you that your manuscript has been deemed suitable for publication in PLOS ONE. Congratulations! Your manuscript is now with our production department. 

Kind regards, 

on behalf of

Dr. Qin Xiang Ng 

Academic Editor

PLOS ONE